# Autonomic Dysfunction Contributes to Impairment of Cerebral Autoregulation in Patients with Epilepsy

**DOI:** 10.3390/jpm11040313

**Published:** 2021-04-17

**Authors:** Shu-Fang Chen, Hsiu-Yung Pan, Chi-Ren Huang, Jyun-Bin Huang, Teng-Yeow Tan, Nai-Ching Chen, Chung-Yao Hsu, Yao-Chung Chuang

**Affiliations:** 1Department of Neurology, Kaohsiung Chang Gung Memorial Hospital, Kaohsiung 83301, Taiwan; fangoe@cgmh.org.tw (S.-F.C.); fornever@cgmh.org.tw (H.-Y.P.); suika68@cgmh.org.tw (C.-R.H.); u9001135@gmail.com (J.-B.H.); tengyeowtan@cgmh.org.tw (T.-Y.T.); naiging@yahoo.com.tw (N.-C.C.); 2College of Medicine, Chang Gung University, Taoyuan 33302, Taiwan; 3Department of Emergency Medicine, Kaohsiung Chang Gung Memorial Hospital, Kaohsiung 83301, Taiwan; 4Department of Neurology, School of Medicine, College of Medicine, Kaohsiung Medical University Hospital, Kaohsiung Medical University, Kaohsiung 80708, Taiwan; cyhsu61@gmail.com; 5Institute for Translation Research in Biomedicine, Kaohsiung Chang Gung Memorial Hospital, Kaohsiung 83301, Taiwan; 6Department of Biological Science, National Sun Yat-sen University, Kaohsiung 80424, Taiwan

**Keywords:** autonomic nervous system dysfunction, cerebral autoregulation, temporal lobe epilepsy, vascular risk, sudden, unexplained death in epilepsy

## Abstract

Patients with epilepsy frequently experience autonomic dysfunction and impaired cerebral autoregulation. The present study investigates autonomic function and cerebral autoregulation in patients with epilepsy to determine whether these factors contribute to impaired autoregulation. A total of 81 patients with epilepsy and 45 healthy controls were evaluated, assessing their sudomotor, cardiovagal, and adrenergic functions using a battery of autonomic nervous system (ANS) function tests, including the deep breathing, Valsalva maneuver, head-up tilting, and Q-sweat tests. Cerebral autoregulation was measured by transcranial Doppler examination during the breath-holding test, the Valsalva maneuver, and the head-up tilting test. Autonomic functions were impaired during the interictal period in patients with epilepsy compared to healthy controls. The three indices of cerebral autoregulation—the breath-holding index (BHI), an autoregulation index calculated in phase II of the Valsalva maneuver (ASI), and cerebrovascular resistance measured in the second minute during the head-up tilting test (CVR_2-min_)—all decreased in patients with epilepsy. ANS dysfunction correlated significantly with impairment of cerebral autoregulation (measured by BHI, ASI, and CVR_2-min_), suggesting that the increased autonomic dysfunction in patients with epilepsy may augment the dysregulation of cerebral blood flow. Long-term epilepsy, a high frequency of seizures, and refractory epilepsy, particularly temporal lobe epilepsy, may contribute to advanced autonomic dysfunction and impaired cerebral autoregulation. These results have implications for therapeutic interventions that aim to correct central autonomic dysfunction and impairment of cerebral autoregulation, particularly in patients at high risk for sudden, unexplained death in epilepsy.

## 1. Introduction

Epilepsy is a common and serious neurological disorder that often leads to morbidity and mortality [1]. The interactions between the autonomic nervous system (ANS) and epileptic seizures are extraordinarily complicated in patients with epilepsy [2,3,4]. In theory, seizures involving the propagation of abnormal neuronal electrical activities may interact with the ANS through central autonomic centers, and may participate in the regulation of autonomic activity [2,4,5]. The effects of epileptic discharge on the ANS through the cortical, limbic, and hypothalamic systems have been suggested [2,4,5]. Clinically, patients with epilepsy frequently manifest autonomic dysfunction, which may involve the cardiovascular, respiratory, gastrointestinal, genial, urinary, and coetaneous systems [2,3,4,6,7]. Autonomic dysfunction in patients with epilepsy is a contributory risk factor for morbidity and mortality due to the potentially fatal effects on the cardiovascular and respiratory systems [2,6,7,8,9,10,11].

The phenomenon of sudden, unexplained death in epilepsy (SUDEP) is a fatal complication of epilepsy associated with uncontrolled seizures [8,12,13]. Autonomic dysregulation, particularly the involvement of cardiovascular function, has been suggested as an important pathophysiological mechanism underlying SUDEP [8,13,14,15,16]. Indeed, central autonomic dysfunction may result in various cardiac dysrhythmias, including tachycardia, bradycardia, and QT prolongation during the interictal and ictal states of epilepsy [6,7,8,17].

Cerebral autoregulation is a mechanism that maintains constant cerebral blood flow despite fluctuations in systemic arterial blood pressure (BP) [18,19]. Evidence shows that cerebral autoregulation disruption occurs in patients with epilepsy, including supranormal demands of cerebral blood flow, disruption of neurovascular coupling, and interictal sympathetic-mediated dysregulation of cerebral blood flow [20,21], occurring under epileptic discharge [22,23,24]. Additionally, some study results have suggested that patients with epilepsy tend to be at higher risk for ischemic stroke [25,26,27,28,29]. As epileptic seizures have potential effects on the hemodynamics of cerebral blood flow, cerebral autoregulation may be an important mechanism by which stable cerebral blood flow is maintained in patients with epilepsy. However, it is unclear whether cerebral autoregulation becomes impaired in patients with epilepsy who have autonomic dysfunction. In our recent study, the central autonomic system was suggested to be potentially perturbed in patients with long-term and severe epilepsy, particularly temporal lobe epilepsy, thereby contributing to autonomic dysfunction and impaired cerebral autoregulation [30]. Therefore, we hypothesize that autonomic dysfunction may contribute to impaired cerebral autoregulation in patients with epilepsy. The present study investigates autonomic function and cerebral autoregulation in patients with epilepsy to determine to what extent these factors may contribute to impaired autoregulation.

## 2. Materials and Methods

This was a prospective case–control study conducted at Kaohsiung Chang Gung Memorial Hospital, Kaohsiung, Taiwan. The Institutional Ethics Committee of Chang Gung Memorial Hospital approved the study protocol (approval number: 98-0805B), and signed informed consent was obtained from all subjects.

### 2.1. Participants

A total of 81 patients (36 females and 45 males) with epilepsy between 18 and 65 years of age were enrolled in the present study through the Epilepsy Outpatient Clinic at Kaohsiung Chang Gung Memorial Hospital between October 2018 and May 2020. Forty-five healthy volunteers (20 females and 25 males) were recruited as normal controls. The mean age of the patients with epilepsy was 35.3 ± 9.70 years, and that for the control group was 37.3 ± 9.89 years.

All enrolled participants were free of diseases that may affect the ANS and cerebral blood flow, including cardiovascular diseases, cerebrovascular diseases, severe traumatic brain injury, polyneuropathies, neurodegenerative diseases, diabetes mellitus, hypertension, endocrine diseases, and autoimmune diseases. Participants taking medications that may affect ANS function were excluded from the present study.

All subjects received neurological evaluations performed by two neurologists, and ANS function was evaluated each morning between 8:00 a.m. and 12:00 noon. Age, gender, body height, body weight, and body mass index (BMI) were recorded. The demographic data of all patients with epilepsy were retrieved from clinical records and interventions, including the duration of epilepsy, etiology, seizure semiology, antiepileptic drug (AED) use, therapeutic state, seizure frequency, and findings of electroencephalography and neuroimaging studies. The seizure types and semiology were classified based on the 2017 recommendations of the International League Against Epilepsy (ILAE) [31].

### 2.2. Evaluation of Autonomic Nervous System Functions

A battery of tests comprising a deep breathing test, the Valsalva maneuver test, the head-up tilt test, and the Q-sweat test was performed to conduct standardized ANS evaluations of the adrenergic, cardiovagal, and sudomotor functions in patients [30,32,33]. Cardiovagal function was assessed by testing the heart rate (HR) response to deep breathing at a defined rate and to the Valsalva maneuver [30,34]. Sudomotor function was measured using a quantitative sudomotor axon reflex test (QSART) that applies the Q-sweat test [30,34,35]. HR was recorded using a basic three-lead electrocardiogram (ECG; Ivy Biomedical, model 101; Branford, CT, USA). Arterial BP was assessed by pulse felt by a professional using beat-to-beat photoplethysmographic records (Finapres BP monitor 2300, Ohmeda; Englewood, CO, USA). BP and ECG were recorded during normal breathing after the patients had undergone a 15 min rest period in the supine position. The sweat output was measured using a Q-sweat device (WR Medical Electronics Co., Stillwater, MN, USA) [30,35].

#### 2.2.1. Deep Breathing Test

A beat-to-beat BP device, ECG, and chest bellows were provided for each subject to undergo the deep breathing test, which was conducted with subjects in a supine position. After a 1 min baseline record, the subjects were assessed for HR variety during six deep breaths within 1 min. The procedure was repeated three times, separated each time by 2 min resting periods. The amplitude of the beat-to-beat variations with respiration were recorded and calculated by the TestWorks software (WR Medical Electronics Co., Maplewood, MN, USA).

#### 2.2.2. Valsalva Maneuver

Each subject repeated the Valsalva maneuver until two responses with similar beat-to-beat BP and HR were obtained. The subjects underwent three repetitions of the Valsalva maneuver test after a 1 min baseline recording. The Valsalva ratio (VR) was calculated from the maximum HR generated by the Valsalva maneuver test divided by the lowest HR within 30 s of peak HR. Beat-to-beat BP was constantly monitored during the Valsalva maneuver using a Finapres monitor and assessed by the TestWorks software.

#### 2.2.3. Head-Up Tilt Test

After producing a 5 min baseline recording with the subject in the supine position, the table was tilted at a 70° angle. Beat-to-beat HR, systolic, diastolic, and mean BP were recorded continuously using Finapres and ECG. Manual BP was obtained 1 min before the head-up tilt test; at 1, 2, 3, and 5 min during the head-up tilt test; and at 1 min post-head-up tilt, as previously described [30].

#### 2.2.4. Quantitative Sudomotor Axon Reflex Test (Q-Sweat Test)

The Q-sweat test protocol, including recording sites, skin preparation, and testing procedure, was performed as previously described [30,35].

#### 2.2.5. Modified Composite Autonomic Scoring Scale (mCASS)

The present study used a composite autonomic scoring scale (CASS) created by the Mayo-QSART [30,33] that was modified (mCASS) and developed in our workshop (using the Q-sweat test instead of a thermoregulatory sweat test as shown in the Appendix A), which served as an indicator of general autonomic function [30,33,35]. The scheme allotted 4 points for adrenergic failure and 3 points each for sudomotor and cardiovagal failure [30,36]. The results of the battery of tests for autonomic function were corrected for the confounding effects of gender and age. The test results were graded semiquantitatively from 0 (no deficit) to 10 (maximal deficit). The mCASS consists of three subscores: sudomotor (0–3), cardiovagal (0–3), and adrenergic (0–4).

### 2.3. Measurements of Cerebral Hemodynamics

The function of cerebral hemodynamic was assessed in a temperature- and humidity-controlled environment using noninvasive transcranial Doppler (TCD) sonography (Multidop XL™, DWL, Sipplingen, Germany) with a headband (Doppler Box, DWL; Compumedics, Charlotte, NC, USA) between 9:00 a.m. and 2:00 p.m. on the same day of ANS evaluation. Beat-to-beat fluctuations in cerebral blood flow velocity and mean cerebral blood flow velocity (CBFV) were measured at the bilateral proximal middle cerebral artery (MCA) using TCD. The MCA was insonated through the temporal window approximately 1 cm above the zygomatic arch at a depth of 35–55 mm using pulsed 2 MHz Doppler probes. The values were monitored continuously during the breath-holding and Valsalva maneuver tests.

#### 2.3.1. The Breath-Holding Index (BHI)

After the subjects rested for 5 min, baseline values of the continuous mean velocity monitored by TCD were collected over 30 s. After normal inspiration, the subjects held their breath for 30 s, and the CBFV of the final 3 s of breath holding were recorded. The procedure was repeated three times, separated each time by 2 min rest periods. The breath-holding index (BHI) was calculated as the percentage increase in CBFV during breath holding, separated by the duration that each subject held their breath [30,37].

#### 2.3.2. An Autoregulatory Index for Phase II (ASI)

After simultaneous performance of the Valsalva maneuver and TCD monitoring, HR, BP, and CBFV were evaluated using the Beatscope Easy software (Finapres Medical Systems B.V., Enschede, The Netherlands). An autoregulatory index for phase II (ASI) was calculated as ASI = (△CBFV/CBFVII) − (△BP/BPII) × 100%. CBFVII and BPII were measured at the beginning of the CBFV recovery slope during phase II [30,38,39]. The differences △CBFV and △BP were calculated for the subsequent 3 s.

#### 2.3.3. Cerebrovascular Resistance (CVR)

The subjects rested for 5 min and then underwent the tilting test with TCD. During the tilt testing with TCD, HR was recorded continuously by 3-lead ECG and beat-to-beat BP was determined for monitoring purposes. Brachial BP was recorded for analysis at 1 min before the tilting test; at 1, 2, 3, and 5 min during tilting; and at 1 min after the tilting test. The estimated mean arterial pressure (MAP) was calculated as ((2 × diastolic BP + systolic BP)/3 and defined as MAP_before_, MAP_1-min_, MAP_2-min_, MAP_3-min_, MAP_5-min_, and MAP_after_ following the above series of time sequences. The mean middle cerebral artery velocity (MCAv) recorded by TCD during the head-up tilting test was defined as MCAv_before_, MCAv_1-min_, MCAv_2-min_, MCAv_3-min_, MCAv_5-min_, and MCAv_after_. Cerebrovascular resistance (CVR) was calculated as MAP/MCAv at the same measured time as described previously [40,41].

### 2.4. Statistical Analysis

All data are presented as the mean ± standard deviation or median (IQR) for continuous variables and as a number (percentage) for categorical variables. The ANS test data, autonomic score (mCASS and sudomotor, adrenergic, and cardiovagal subscores), serial HR, MAP, MCAv, and cerebral hemodynamic index (BHI, ASI, and CVR) between patients with epilepsy and the controls were calculated using the Mann–Whitney U test or Fisher’s exact test. Correlations between the autonomic scores (mCASS and sudomotor, adrenergic, and cardiovagal subscores) and the cerebral hemodynamic index (BHI, ASI, and CVR) were assessed using Spearman’s rho correlation analysis. The Kruskal–Wallis test was used to analyze the differences between the cerebral hemodynamic indices in the controls and in the epileptic subtypes. A value of *p* < 0.05 was established as having statistical significance. All statistical analyses were performed using IBM SPSS Statistics Version 22.0 (IBM, Armonk, NY, USA).

## 3. Results

### 3.1. Demographic and Clinical Characteristics

No significant differences were found for gender and age between patients and the control subjects. The BMI values in the patients with epilepsy were slightly higher than that in controls, but this difference was not statistically significant (24.2 ± 4.66 vs. 22.7 ± 2.29; *p* = 0.197). The baseline demographic and clinical characteristics of the 81 patients with epilepsy are shown in Table 1.

### 3.2. Impairment of Autonomic Function and Cerebral Autoregulation in Patients with Epilepsy

The results of the autonomic scores (mCASS and sudomotor, adrenergic, and cardiovagal subscores) in the patients with epilepsy and the controls are shown in Table 2. The autonomic function scores, including those for adrenergic, cardiovagal, and sudomotor function and for mCASS, were significantly higher in patients with epilepsy than in the controls.

Additionally, three cerebral autoregulation parameters were tested in the patient and control groups: BHI, ASI, and CVR. As we observed previously [30], BHI and ASI were significantly lower in patients with epilepsy than in the control group (Table 3). In a series of head-up tilting tests, calculation of CVR from the recorded MAP and MCAv revealed that the patients with epilepsy had a significant decrease in the CVR index compared with the control group in for CVR_2-min_ during the head-up tilting test, as shown in the Appendix A. However, the CVR at the other time point did not show a significant difference between epilepsy patients and the control group. A lower CVR_2-min_ (median = 1.373) was noted in patients with epilepsy compared with CVR_2-min_ (median = 1.544) in the controls (Table 3).

From the autonomic questionnaires, we found that 9 patients had undergone syncope and that 47 experienced orthostatic hypotension. Using the Mann–Whitney U test to compare the autonomic function scores (mCASS and sudomotor, adrenergic, and cardiovagal subscores) and the cerebral autoregulation index (BHI and ASI), no significant difference was found between syncope and non-syncope patients or between patients with orthostatic hypotension and non-orthostatic hypotension. However, the cerebral autoregulation index, CVR_2-min_, showed statistically significant differences between patients with orthostatic hypotension and non-orthostatic hypotension. In the non-orthostatic hypotension group, CVR_2-min_ was 1.28 (1.12, 1.42) mmHg × s/cm (median (IQR)), and in the orthostatic hypotension group, CVR_2-min_ was 1.40 (1.26, 1.55) mmHg × s/cm; *p* = 0.019.

### 3.3. Autonomic Dysfunction and Impaired Cerebral Autoregulation in Patients with Epilepsy

Table 4 lists the correlation analysis for autonomic function and cerebral autoregulation parameters and the semiology and demographic variables in patients with epilepsy. Adrenergic function and the cerebral autoregulation index (CVR_2-min_) showed significant correlation with age in patients with epilepsy. Male patients had a lower CVR_2-min_ compared with females. Patients with chronic epilepsy and who experienced a longer duration of epilepsy are prone to have autonomic dysfunction (higher mCASS) and cerebral dysregulation (lower ASI index). Patients with refractory epilepsy and a high frequency of seizures had positive correlations with mCASS, and cardiovagal and sudomotor subscores. Patients using AED had no significant differences.

### 3.4. Autonomic Dysfunction Correlated with Impaired Cerebral Autoregulation in Patients with Epilepsy

The results of correlation analysis between the autonomic function scores (mCASS and adrenergic, cardiovagal, and sudomotor subscores) and the cerebral hemodynamic parameters (BHI, ASI, and CVR_2-min_) are shown in Table 5. BHI and ASI had significantly negative correlations with all autonomic scores (mCASS and adrenergic, cardiovagal, and sudomotor subscores). The lower CVR_2-min_ index correlated significantly with higher mCASS, and adrenergic and sudomotor subscores.

### 3.5. Impaired Cerebral Autoregulation in Different Forms of Epilepsy

To determine the extent of autonomic dysfunction and dysregulation of cerebral blood flow, the patients were divided by epileptic type into temporal lobe epilepsy (female (*n*)/male (*n*) = 19/14; age = 38.2 ± 10.87 years), extratemporal epilepsy (female (*n*)/male (*n*) = 7/20; age = 34.5 ± 8.56 years), and generalized epilepsy (female (*n*)/male (*n*) = 10/11; age = 31.5 ± 7.87 years). No statistically significant differences were found in sex and age. In the autonomic function parameters, including mCASS and adrenergic, cardiovagal, and sudomotor subscores, the three forms of epilepsy showed significant impairment compared to the controls. Although no significant differences were found between temporal lobe epilepsy, extratemporal lobe epilepsy, and generalized epilepsy, the indices for cerebral autoregulation, ASI, and CVR_2-min_ in the three forms of epilepsy were significantly lower than those in the controls (Table 6). Since BHI in the three forms of epilepsy was significantly lower than that in the controls, BHI in patients with temporal lobe epilepsy was statistically significant (*p* < 0.05) compared to BHI in the other groups, suggesting that temporal lobe epilepsy is associated with worse cerebral autoregulation than the other two forms (Table 6). There are 33 patients with temporal lobe epilepsy, including 19 with left and 14 with right temporal lobe epilepsy. In a comparison of the autonomic function scores (mCASS and adrenergic, cardiovagal, sudomotor subscores) and the cerebral autoregulation indices (BHI, ASI, and CVR_2-min_) between right and left lateralized temporal lobe epilepsy using the Mann–Whitney U test, the results showed no significant difference between autonomic function and cerebral autoregulation (*p* > 0.005).

## 4. Discussion

The results of the present study confirm that patients with epilepsy frequently experience ANS dysfunction and impaired cerebral autoregulation. Using a battery of tests [30,32,33] consisting of a deep breathing test, the Valsalva maneuver test, the head-up tilt test, and the Q-sweat test, patients with epilepsy were shown to experience a higher degree of autonomic dysfunction than the controls. Using three cerebral autoregulation indices measured by a TCD monitor, including BHI, ASI, and CVR_2-min_, cerebral autoregulation was shown to be more significantly impaired in patients with epilepsy than in the control subjects. Interestingly, lower BHI, ASI, and CVR_2-min_ values in the patients with epilepsy correlated negatively with all autonomic scores, strongly suggesting that autonomic dysfunction contributes to impaired cerebral autoregulation in patients with epilepsy.

Autonomic dysfunction is common in a range of neurological disorders, including epilepsy [4,42], multiple sclerosis [43], Alzheimer’s disease [44], Parkinson’s disease, Shy–Drager syndrome [45,46,47], dementia with Lewy bodies [45,48], and multiple system atrophy [45,48]. Epilepsy is a heterogeneous condition with varied etiologies, including in genetics, vascular insults, infection, trauma, neoplasms, and toxic exposures [49]. However, epileptic discharges that act on ANS function are considered involved in the neural circuit activities of several different stress-responsive brain regions, including the prefrontal cortex, amygdala, and hippocampus and in the hypothalamic–pituitary–adrenal (HPA) neuroendocrine system [2,4,5,49]. Epileptic seizures that are not involved in neural circuit activity but involved in the neuroendocrine systems are thought to be prone to having ANS dysfunction in patients with epilepsy [49]. Our previous study [30] showed that perturbed brain-derived neurotrophic factor and insulin-like growth factor 1 signaling in the central autonomic system contribute to autonomic dysfunction and impaired cerebral autoregulation in patients with focal epilepsy.

In the present study, autonomic dysfunction and dysregulation of cerebral blood flow were present in patients with advanced age and were more frequent in female patients than in male patients. In healthy adults, autonomic function and cerebral autoregulation have been notably decreased with age and in females [50,51]. According to seizure semiology, patients with chronic epilepsy and a longer duration of epilepsy are more prone to have autonomic dysfunction (higher mCASS) and poor cerebral regulation (lower ASI index). In patients with refractory epilepsy with a high frequency of seizures, autonomic dysfunction is more significant compared with patients with well-controlled epilepsy. However, AED therapy did not result in significant differences between the types of epilepsy in the present study, which may be due to the small number of patients and the complexity of AED therapy.

In our previous study [30], we selected the indices for evaluation of cerebral autoregulation, including BHI and ASI. Additionally, we used CVR_2-min_ to evaluate the cerebral autoregulation in the present study. In head-up tilt testing, the classic orthostatic hypotension caused by autonomic failure in increasing systemic vascular resistance, which results in pooling of blood at the lower extremities, but while standing, it may take from 30 s to 3 min for BP to fall [41,52]. Evidence also has shown that changes in systemic hemodynamics and in cerebral blood velocity indicate that CVR integrates cerebral autoregulation with an impending syncope [41,52]. In the present study, we found that patients with epilepsy have a significant decrease in CVR index compared to the controls in the 2 min (between 30 s to 3 min) during the head-up tilting test. Thus, CVR_2-min_ may be a novel index for the evaluation of cerebral autoregulation. Tagged together with BHI and ASI, patients with epilepsy who have impaired autonomic function are also susceptible to decreasing cerebral autoregulation.

Importantly, the results of the correlation analysis between autonomic dysfunction in patients with epilepsy show high correlation with the cerebral hemodynamic parameters (BHI, ASI, and CVR_2-min_). This phenomenon indicates that the increased degree of autonomic dysfunction in patients with epilepsy may augment the dysregulation of cerebral blood flow. In the present study, the cerebral autoregulation index, CVR_2-min_, is significantly lower in patients who experienced orthostatic hypotension compared to that in patients without orthostatic hypotension. These results suggest that patients with epilepsy who have clinical signs of autonomic dysfunction, such as orthostatic hypotension, may present impaired cerebral autoregulation. The autonomic dysfunction in patients with epilepsy is a contributory risk factor for morbidity and mortality due to the potential effects on dysfunction of cerebral upregulation that are possibly involved in the regulation of cardiovascular systems [2,6,7,8,9,10,11,16].

Temporal lobe epilepsy is one of the most common forms of focal onset epilepsy. Autonomic dysfunction and dysregulation of cerebral blood flow are often present in patients with temporal lobe epilepsy [42,53,54,55]. In temporal lobe epilepsy, interictal or ictal seizure activities that originated from a temporal focus may disrupt the limbic circuitry and HPA axis regulation in the central autonomic system, which contributes to dysfunction of the ANS and impaired cerebral autoregulation in patients with temporal lobe epilepsy [55,56]. In addition, regional uncoupling of blood flow and metabolism in the temporal area indicating temporal lobe epilepsy has different underlying mechanisms of regulation between brain metabolism and cerebral blood flow in the brain [57]. Perhaps autonomic dysfunction may lead to impaired cerebral autoregulation in patients with epilepsy and may increase the risk of SUDEP.

To evaluate whether different forms of epilepsy have different extents of autonomic dysfunction and cerebral dysregulation, the patients in the present study were divided into three forms of epilepsy for analysis according to their focus of epileptogenesis: temporal lobe epilepsy, extratemporal epilepsy, and generalized epilepsy. Among the autonomic function parameters, all three forms of epilepsy showed significant impairment compared to the controls. No significant differences were found between the three forms of epilepsy. However, the index of cerebral autoregulation, BHI, in patients with temporal lobe epilepsy was statistically significantly (*p* < 0.005) different compared with the other groups, which may indicate that patients with temporal lobe epilepsy may have worse cerebral autoregulation than those with other forms of epilepsy. Recently, Dono et al. [53] reported that the patients with temporal lobe epilepsy exhibited abnormalities in the regulation of autonomic cardiovascular functions. Patients with left temporal lobe epilepsy may have a lower risk for the development of cardiac dysfunction and may be less susceptible to the development of SUDEP. However, in the present study, we found significant differences in autonomic function and cerebral autoregulation between patients with left and right temporal lobe epilepsy. These results may be related to the small number of study patients. Further studies with a larger sample and longer follow-up are necessary to determine whether patients with temporal lobe epilepsy have a higher incidence of autonomic dysfunction and impaired cerebral autoregulation, or an increased risk of SUDEP.

## 5. Conclusions

The results of the present study show that autonomic dysfunction and impaired cerebral autoregulation are present in patients with epilepsy, providing novel insight into clinical observations regarding epilepsy. Long-term epilepsy, a high frequency of seizures, and refractory epilepsy, particularly temporal lobe epilepsy, may contribute to advanced autonomic dysfunction and, subsequently, to impaired cerebral autoregulation in patients with epilepsy. These results are important for our understanding of the pathogenesis of autonomic dysfunction and dysregulation of cerebral blood flow in patients with epilepsy and have implications for therapeutic interventions that aim to correct central autonomic dysfunction and impairment of cerebral autoregulation, particularly in patients at high risk for SUDEP.

## Figures and Tables

**Table 1 jpm-11-00313-t001:** Demographic and clinical characteristics of 81 patients with epilepsy.

**Age**	35.3 ± 9.70
**Female**	34.2 ± 8.32
**Male**	36.1 ± 10.70
**Sex (Female/Male), n (%)**	36 (44.4%)/45 (55.6%)
**Duration of Epilepsy (Years)**	17.8 ± 9.31
**Form of Epilepsy, n (%)**	
Temporal lobe epilepsy	33 (40.7%)
Extratemporal lobe epilepsy	27 (33.3%)
Generalized epilepsy	21 (25.9%)
Seizure frequency (per month)	2.0 ± 5.2
**Seizure Control, n (%)**	
Refractory epilepsy	30 (37.0%)
Seizure free *	29 (35.8%)
**AED Numbers, n (%)**	
Single	25 (30.9%)
Multiple	56 (69.1%)

The values are expressed as mean ± standard deviation. * Defined as seizure free for more than 12 months under current AED therapy. Abbreviations: AED, antiepileptic drug.

**Table 2 jpm-11-00313-t002:** Comparison of autonomic function parameters in patients with epilepsy and the controls.

	Score	Patients with Epilepsy (*n* = 81)	Controls (*n* = 45)	*p* Value
Adrenergic	0	16	45	<0.001 *
	1	56	0	
	2	9	0	
Cardiovagal	0	39	45	<0.001 *
	1	27	0	
	2	12	0	
	3	3	0	
Sudomotor	0	14	45	<0.001 *
	1	40	0	
	2	12	0	
	3	15	0	
mCASS	0	3	45	<0.001 *
	1	8	0	
	2	19	0	
	3	23	0	
	4	15	0	
	5	10	0	
	6	3	0	

Abbreviations: mCASS, modified composite autonomic severity score. * Significant difference between patients with epilepsy and controls using Fisher’s exact test.

**Table 3 jpm-11-00313-t003:** Comparison of cerebral autoregulation parameters in patients with epilepsy and controls.

	Patients with Epilepsy (*n* = 81)	Controls (*n* = 45)	*p* Value
BHI	0.88 (0.64, 1.08)	1.05 (0.85, 1.37)	0.001 *
ASI	−1.84 (−5.42, 2.99)	8.26 (3.12, 10.22)	0.001 *
CVR_2-min_	1.35 (1.20, 1.46)	1.54 (1.26, 2.06)	0.004 *

The values are presented as autonomic function scores and as median (IQR). * Significant difference between epilepsy patients and controls using the Mann–Whitney U test. Abbreviations: BHI, breath-holding index of middle cerebral artery; ASI, a second autoregulation index for phase II in the Valsalva maneuver test; CVR_2-min_, cerebrovascular resistance in the second minute of the head-up tilting test.

**Table 4 jpm-11-00313-t004:** Correlation analysis of autonomic function, cerebral autoregulation parameters, and demographic variables in patients with epilepsy and in the controls.

	Adrenergic	Cardiovagal	Sudomotor	mCASS	BHI	ASI	CVR_2-min_
Age (years)	0.219 (0.049 *)	0.115 (0.309)	0.055 (0.628)	0.189 (0.091)	−0.158 (0.160)	0.073 (0.546)	0.321 (0.005 *)
Sex (female/male)	−0.146 (0.192)	−0.024 (0.830)	−0.075 (0.507)	−0.075 (0.506)	−0.025 (0.825)	0.096 (0.428)	0.278 (0.017 *)
Duration of epilepsy (years)	0.179 (0.112)	0.047 (0.680)	0.188 (0.095)	0.239 (0.033 *)	−0.127 (0.263)	0.243 (0.044 *)	0.182 (0.123)
Seizure frequency (per month)	0.124 (0.271)	0.219 (0.050 *)	0.205 (0.066)	0.339 (0.002 *)	−0.121 (0.283)	0.011 (0.925)	0.074 (0.532)
Seizure control							
Refractory epilepsy	0.069 (0.539)	0.199 (0.074)	0.229 (0.040 *)	0.321 (0.004 *)	−0.049 (0.663)	0.026 (0.829)	0.126 (0.283)
Seizure free #	−0.165 (0.142)	−0.185 (0.099)	−0.213 (0.056)	−0.333 (0.002 *)	0.116 (0.304)	−0.055 (0.652)	−0.119 (0.313)
AED numbers							
Single	0.098 (0.382)	0.043 (0.704)	−0.141 (0.210)	−0.063 (0.575)	−0.030 (0.792)	−0.093 (0.442)	−0.137 (0.244)
Multiple	−0.098 (0.382)	−0.043 (0.704)	0.141 (0.210)	0.063 (0.575)	0.030 (0.792)	0.093 (0.442)	0.137 (0.244)

The values are expressed as correlation coefficients (σ) and significance (*p*) using Spearman’s rho correlation. # Defined as seizure free for more than 12 months under current AED therapy. * Correlation is significant at the 0.05 level (2-tailed).

**Table 5 jpm-11-00313-t005:** Correlation analysis between autonomic function parameters and cerebral autoregulation parameters in patients with epilepsy.

	Adrenergic	Cardiovagal	Sudomotor	mCASS
BHI	−0.285 (0.001 *)	−0.198 (0.026 *)	−0.246 (0.005 *)	−0.289 (0.001 *)
ASI	−0.444 (0.001 *)	−0.388 (0.001 *)	−0.351 (0.001 *)	−0.455 (0.001 *)
CVR_2-min_	−0.224 (0.014 *)	−0.088 (0.343)	−0.200 (0.029 *)	−0.214 (0.019 *)

The values are expressed as correlation coefficients (σ) and significance (*p*). * Statistical significance (*p* < 0.05) in Spearman’s rho correlation analysis. Abbreviations: BHI, breath-holding index of middle cerebral artery; ASI, a second autoregulation index for phase II in the Valsalva maneuver test; CVR_2-min_, cerebrovascular resistance in the second minute of the head-up tilting test.

**Table 6 jpm-11-00313-t006:** Comparison of cerebral hemodynamics in the three different forms of epilepsy and in the controls.

	Temporal Lobe Epilepsy (*n* = 33)	Extratemporal Lobe Epilepsy (*n* = 27)	Generalized Epilepsy (*n* = 21)	Controls (*n* = 45)	*p* Value
BHI	0.80 (0.49, 0.98) ^#^	0.92 (0.65, 1.10)	0.96 (0.63, 1.24)	1.05 (0.85, 1.37)	0.001 *
ASI	−1.12 (−3.76, 4.26)	−2.51 (−5.40, −0.42)	−2.95 (−9.18, 3.12)	8.26 (3.12, 10.22)	0.001 *
CVR_2-min_	1.29 (1.18, 1.45)	1.39 (1.19, 1.55)	1.38 (1.24, 1.44)	1.54 (1.26, 2.06)	0.036 *

The values are presented as medians (IQR). * Statistically significant (*p* < 0.05) differences between the patients with different forms of epilepsy and the controls using the Kruskal–Wallis test. ^#^ The results after multiple comparisons indicate that the breath-holding indices (BHI) in patients with temporal lobe epilepsy are statistically significant (*p* < 0.005) compared with BHI in other groups. Abbreviations: BHI, breath-holding index of middle cerebral artery [30,37]; ASI, a second autoregulation index for phase II in the Valsalva maneuver test [30,38,39]; CVR_2-min_, cerebrovascular resistance in the second minute of the head-up tilting test [40,41].

## Data Availability

The data used to support the findings of this study are included within the article.

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
