# Peer review of "Autonomic Dysfunction Contributes to Impairment of Cerebral Autoregulation in Patients with Epilepsy"

_jpm, 2021, doi:10.3390/jpm11040313_

Round 1

Reviewer 1 Report

Dear authors,

I would like to congratulate you with very interesting research and very well written article. Below are some of my questions and comments. 

1. I suggest adding the missing reference to Table 6 in Section 3.5.

2. Your finding is “The patients with epilepsy had a significant decrease in CVR index compared with the control group in the CVR2-min during the head-up tilting test." (Table 3).
I suggest providing CVR values for other time points (before the tilting test, 1,3 and 5 min during tilting, and 1 min after the tilting test), although, according to you, they did not differ between patients and controls. In order not to overload the article, it can be submitted as supplemental material.

3. Your finding is “Autonomic functions were impaired during the interictal period in patients with epilepsy compared with healthy controls.”
My question: was there a minimum time between the last epileptic seizure and the assessment of ANS function in the patient group?

4. For discussions: Dono F et al. (2020) recently reported that the patients with temporal lobe epilepsy (TLE) exhibit a lateralized cardiac autonomic control. Left TLE patients may have a lower risk of developing cardiac dysfunctions and less susceptible to develop SUDEP (Front Neurol 2020 Aug 14;11:842. doi: 10.3389/fneur.2020.00842).

My question: have you tried to divide your patients into subgroups by TLE lateralization and to compare the results of ANS and CVR dysfunction?

Thank you.

Author Response

Response to Reviewer #1

English have been editing (English editing ID: English-28983 MDPI manuscript ID: jpm-1173226)

  1. I suggest adding the missing reference to Table 6 in Section 3.5.

Response: We have added the reference in Section 3.5. (Table 6, Line 325-327)

  1. Your finding is “The patients with epilepsy had a significant decrease in CVR index compared with the control group in the CVR2-min during the head-up tilting test." (Table 3).

I suggest providing CVR values for other time points (before the tilting test, 1,3 and 5 min during tilting, and 1 min after the tilting test), although, according to you, they did not differ between patients and controls. In order not to overload the article, it can be submitted as supplemental material.

Response: Thank you for your suggestions. We have added the information of CVR values in the supplemental materials (Table S2).

  1. Your finding is “Autonomic functions were impaired during the interictal period in patients with epilepsy compared with healthy controls.”

My question: was there a minimum time between the last epileptic seizure and the assessment of ANS function in the patient group?

Response: In the present study, all enrolled patients with epilepsy who received ANS evaluation were stationary, and tests were done in the interictal stage. We did not monitor the minimum time between the last epileptic seizure and the assessment of ANS function.

  1. For discussions: Dono F et al. (2020) recently reported that the patients with temporal lobe epilepsy (TLE) exhibit a lateralized cardiac autonomic control. Left TLE patients may have a lower risk of developing cardiac dysfunctions and less susceptible to develop SUDEP (Front Neurol 2020 Aug 14;11:842. doi: 10.3389/fneur.2020.00842).

My question: have you tried to divide your patients into subgroups by TLE lateralization and to compare the results of ANS and CVR dysfunction?

Response: In the present study, 33 patients with temporal lobe epilepsy, including 19 with left and 14 with right temporal lobe epilepsy. Comparison of the autonomic function scores (mCASS and adrenergic, cardiovagal, sudomotor subscores) and the cerebral autoregulation indexes (BHI, ASI and CVR2min) between right and left lateralized temporal lobe epilepsy by using the Mann-Whitney U test, the results showed no significant difference in autonomic functions and cerebral autoregulation (p > 0.005). These results may be related with the small number of study patients. We have provided our results in Results (Line 307-312) and elaborated Discussion (Line 410 to 417)

Reviewer 2 Report

In this study the authors show that autonomic dysfunction and impaired cerebral autoregulation are present in patients with epilepsy. The study is interesting and well designed, further exploring the link between EP, ANS dysfunction and possibly SUDEP.

I have a few comments:

  1. The authors state that subjects taking medications affecting ANS were excluded. However, could the anti-EP medications have influenced the results in the EP Group, especially regarding cerebral autoregulation?
  2. Did any of the EP patients have documented episodes of syncope (i.e. TLOC not caused by EP) or any clinical signs of autonomic dysfunction such as OH?
  3. Were there any differences in hemodynamic parameters during TILT between EP patients and controls?
  4. The authors conclude that their results "have implications for therapeutic interventions to correct central autonomic dysfunction and impairment of cerebral autoregulation, particularly in older adult patients and individuals at high risk for SUDEP".
    Firstly, it would be interesting to hear the authors opinions on how such interventions may be done. Secondly, the study involves relatively Young subjects and not older subjects?

Author Response

Response to reviewer #2

  1. The authors state that subjects taking medications affecting ANS were excluded. However, could the anti-EP medications have influenced the results in the EP Group, especially regarding cerebral autoregulation?

Response: In the patients with epilepsy who received anti-epileptic drug (AED) therapy always complicated. We try to investigate which AED related to impaired ANS function, but patients using AED had no significant differences. AED therapy did not show significant differences between the types of epilepsy in the present study, which may be due to the small number of patients and the complexity of AED therapy (Result: Line 270; Discussion: Line. 362-364).

  1. Did any of the EP patients have documented episodes of syncope (i.e. TLOC not caused by EP) or any clinical signs of autonomic dysfunction such as OH?

Response: Thank you for your suggestion. We have use questionnaire to address the clinical signs. We have added our information and results in Results (Line 244-253) and Discussion (Line 382-386).

  1. Were there any differences in hemodynamic parameters during TILT between EP patients and controls?

Response We have added these parameters in in the supplemental materials (Table S2)

  1. The authors conclude that their results "have implications for therapeutic interventions to correct central autonomic dysfunction and impairment of cerebral autoregulation, particularly in older adult patients and individuals at high risk for SUDEP".

Firstly, it would be interesting to hear the authors opinions on how such interventions may be done. Secondly, the study involves relatively Young subjects and not older subjects?

Response: Thank you for your comments. We change the conclusion to “These results have implications for therapeutic interventions to correct central autonomic dysfunction and impairment of cerebral autoregulation, particularly in patients at high risk for sudden unexplained death in epilepsy.”

 English has bee editing (English editing ID: English-28983 MDPI manuscript ID: jpm-1173226)